# Improvement in the Pharmacological Profile of Copper Biological Active Complexes by Their Incorporation into Organic or Inorganic Matrix

**DOI:** 10.3390/molecules25245830

**Published:** 2020-12-10

**Authors:** Mihaela Badea, Valentina Uivarosi, Rodica Olar

**Affiliations:** 1Department of Inorganic Chemistry, Faculty of Chemistry, University of Bucharest, 90-92 Panduri Str., 050663 Bucharest, Romania; mihaela.badea@chimie.unibuc.ro; 2Department of General and Inorganic Chemistry, Faculty of Pharmacy, Carol Davila University of Medicine and Pharmacy, 6 Traian Vuia Str., 020956 Bucharest, Romania

**Keywords:** complex, copper(II), matrix, formulation, antimicrobial activity, biofilm

## Abstract

Every year, more Cu(II) complexes are proven to be biologically active species, but very few are developed as drugs or entered in clinical trials. This is due to their poor water solubility and lipophilicity, low stability as well as in vivo inactivation. The possibility to improve their pharmacological and/or oral administration profile by incorporation into inorganic or organic matrix was studied. Most of them are either physically encapsulated or conjugated to the matrix via a moiety able to coordinate Cu(II). As a result, a large variety of species were developed as delivery carriers. The organic carriers include liposomes, synthetic or natural polymers or dendrimers, while the inorganic ones are based on carbon nanotubes, hydrotalcite and silica. Some hybrid organic-inorganic materials based on alginate-carbonate, gold-PEG and magnetic mesoporous silica-Schiff base were also developed for this purpose.

## 1. Introduction

Nowadays, the aspects concerning the use of metallodrugs as an alternative for organic molecules is an important aspect of modern medicine, and some of the metal complexes that are rapidly gaining credit in this field are those of copper. The use of copper for this purpose can be traced to ancient Egypt, where this metal was used for water sterilization, while its compounds were used for wounds disinfection in the chest area. Additionally, a mixture of copper oxide and honey/rose oil was used in antiquity for wound disinfection and deworming. The powder malachite (copper hydroxide carbonate, Cu_2_CO_3_(OH)_2_) and copper oxide were used for ophthalmic conditions in ancient Persia [1].

The effect of copper on the immune system was first mentioned in 1867, during the cholera epidemics of that period in France, when it was noted that people working in copper mines did not get sick. Later, in 1939, the therapeutic potential of copper in the treatment of rheumatic diseases was discovered when it was observed that Finnish miners from copper exploitations did not suffer from rheumatism [1].

Nowadays there are some copper compounds used as drugs or supplements either to manage the neurological disorders in Menkes disease or for the treatment of some conditions based on its ability to coordinate and interact with a wide range of biomolecules. Among them, copper acetate [Cu_2_(CH_3_COO)_4_(OH_2_)_2_] (**1**) and histidinate [Cu(His)_2_] (**2**) (HHis is histidine) are used in the intravenous (i.v.) administration for relieving symptoms in Menkes disease [2] (Figure 1). Copper sulphate and gluconate [Cu(glu)_2_] (**3**) (HGlu is gluconic acid) are usually found in the complex medication of iron-deficiency anemia [3,4], while chlorophyllin (**4**) in pills, spray or ointment formulations is recommended for wound odor elimination, radiation burns, inflammatory diseases and even for liver cancer prevention [5,6,7]. The mammalian Cu, Zn-superoxide dismutase (SOD1 oxidized active site (**5**)) formulated as injection or cream is recommended for inflammatory diseases. This enzyme is also recommended in diabetic complications, atherosclerosis, Alzheimer’s disease, cancer as well as in rheumatic arthritis [8,9,10,11,12]. Some ointments containing Cu(II) and Zn(II) glycinates, such as [Cu(Gly)] (HGly is glycine) (**6**), are recommended for skin conditions [13].

Moreover, [Cu(atsm)] (**7**) (H_2_atsm is diacetylbis(N-4-methylthiosemicarbazone) as a drug for amyotrophic lateral sclerosis (ALS) [14], [Cu(dmbpy)(acac)(OH_2_)]NO_3_ (dmbpy is 4,4-dimethyl-2,2′-bipyridine, Hacac is acetylacetone) (**8**) from the Casíopeinas^®^ family [15] as well as (**4**) combined with disulfiram (dsf) [16] as antitumor drugs are currently underway in clinical trials.

Copper was selected for medical purposes because of: (i) its essential character, (ii) the borderline behavior as a Lewis acid, (iii) the stereochemical versatility, (iv) the ability to change its oxidation state and (v) the relative low systemic toxicity. This is an essential element found in traces in the human body at a level of 0.11 g/70 kg. As component of copper-proteins, it plays an important role in many redox processes as a result of easy oxidation state changing between +1 and +2. Among them, the copper-enzymes are involved in iron homeostasis, superoxide disproportionation, melanin, catecholamines and neuropeptide synthesis as well as collagen with elastin cross-linking [17,18,19,20].

In the human body, Cu(II) can interact with a wide range of biomolecules as a result of its ability to coordinate both hard and soft ligands. The interaction with DNA and proteins is exploited in order to develop some valuable antitumor, anti-inflammatory and antimicrobial species [17,18,19,20]. On the other hand, the interaction with cytosolic peptides glutathione and metallothioneins can inactivate the Cu(II) complexes [21]. The Jahn Teller effect that operate for Cu(II) complexes together with ligand properties are responsible for the wide range of known stereochemistries for this ion. The ability to easily change the stereochemistry explains the interaction with different biomolecules, an interaction that can also be assisted by the ligand [17,18,19,20].

Due to its redox properties, copper is involved both in the regulation of physiological processes and in the generation of highly reactive oxygen species (ROS), both responsible for its antitumor activity. The Cu(II) complexes with an appropriate potential are reduced by superoxide anion to Cu(I) species that reduce H_2_O_2_ resulted by SOD1 activity to HO, radical responsive by DNA strands splitting [22].

The copper homeostasis is well-tuned, and as a result, any copper excess is stored as a complex with metallothioneins [17,18,19,20,23]. This metal ion capture provides a way to reduce the complexes’ toxicity. However, the copper toxicity is manifested in Wilson disease, where the organism escapes the normal metabolic pathways and allows the metal accumulation in the liver [17]. Wilson’s disease is manifested by neurological and liver damages and the appearance of a characteristic brown ring in the periphery of the cornea (“Kayser–Fleischer ring”), accompanied by increased copper levels in urine and the affected brain [24]. The non-Wilson’s disease disorders (copper toxicosis), such as Indian childhood cirrhosis (ICC), Idiopathic copper toxicosis and endemic Tyrolean infantile cirrhosis, appear to be caused by excessive accumulation of copper uptake through food and drinking water, although a genetic predisposition has also been linked to ICC-like illness [25]. Excessive copper intake can cause a metallic taste, salivation nausea, vomiting, abdominal pain and cramps, headache, dizziness, weakness and diarrhea. High uptakes of copper may cause liver and kidney damage, neurodegenerative disorders and even death [26]. Toxic concentrations of copper can occur when copper ingestion is over 1 g (usually taken in suicide attempts) and the estimated lethal dose in an untreated adult is 10–20 g. Systemic toxicity occurs at serum copper levels of 78.5 μmol/L [27]. The human body makes contact with a large variety of sources of cooper, including copper water pipes, copper cookware, drinking water, foods, birth control pills and Cu intrauterine devices, vitamin and minerals supplements, fungicides and added copper in swimming pools [25,28]. These concerns regarding copper toxicity must be taken into account for biologically active copper complexes.

Many copper complexes were designed as versatile and reliable tools for developing biologically active compounds. The ligand structure and ability to both enhance the stability and regulate the non-covalent interactions with biomolecules were selected in order to finely tune the complexes’ desirable properties [29]. As a result, many copper complexes with potential therapeutic activities were developed. The biological application of these complexes extends from enzyme inhibition or species with enzyme-like activity [30,31,32,33] to active species with antimicrobial, anti-inflammatory or antitumor activity [17,18,19,20,22,23,29,34,35,36,37,38,39,40,41], aspects presented in several reviews. 

However, a small number of copper-based drugs are in current use as a result of reduced stability both in water and acidic environment and an unfavorable balance between water solubility and lipophilicity. In order to solve these problems, some delivery carriers have been developed to increase both the drug protection and efficacy. These systems were developed based on the data obtained during the attempts to improve the similar issues related to platinum-based drugs. As a result, the copper systems were combined with a wide range of organic, inorganic or hybrid (organic-inorganic)-based carriers.

## 2. Delivery Systems for Copper Active Complexes

### 2.1. Organic-Based Carriers

Biologically active copper complexes (bacc) administrated through a delivery system are protected against rapid metabolism in the presence of hydrochloric acid, water or serum and cytoplasmic proteins before the interaction with the target biomolecules. Moreover, such systems enhance the half-life of active compound in the blood stream and increase its cellular uptake by endocytosis, especially in tumor tissues, which are characterized by an Enhanced Permeability and Retention effect (EPR effect) [42]. 

Based on the experience achieved in platinum-based drugs encapsulation, several bacc were either physically encapsulated into a functionalized organic or inorganic matrix or conjugated to its backbone via a multichelate coordinative site. The matrix can be a synthetic one, selected from what is usually tested as biocompatible and biodegradable and already used in pharmaceutical formulations, or can be a natural polysaccharide or protein system.

Physical encapsulation is rarely explored for Cu(II) complexes, since it depends on the both the solubility and polarity of the compound. Only a few delivery systems, where the bacc is not conjugated to a matrix, have been reported so far. Besides polymeric systems, bacc encapsulation into liposomes also seems to be a good strategy in the field. The reported data indicate that the polymer–drug conjugation is by far the preferred approach, since the chelate coordination enhances the stability, and as a result, the circulating time of active compounds in the body [43]. For such systems, the type of polymer, its functionalization as well as properties and structure of complexes are extremely important factors in order to achieve the desired properties. 

#### 2.1.1. Liposomal Systems

Liposomes are spherical vesicles with an aqueous core surrounded by a lipid bilayer similar in morphology as natural phospholipid membranes. As a result, these systems are suitable to encapsulate both hydrophilic species in the aqueous core and hydrophobic ones into the lipid bilayer [44]. 

Based on the experience gained in encapsulation of other inorganic species, some liposomal formulations were developed for bacc, in order to improve their water solubility and deliverability rate as presented in Table 1. Thus, dimyristoyl phosphatidylcholine (DMPC)-based long circulating nanoliposomes containing [Cu(phen)Cl_2_] (phen is 1,10-phenantroline) (**9**), a potent aquaporin inhibitor, were developed and assayed against several melanoma and colon cancer cells. All cell lines were sensitive with a half-inhibitory concentration (IC_50_) value around 3 μM. Furthermore, the in vivo studies revealed no toxic effects after parenteral administration [45]. The same complex in another nanoliposomal formulation demonstrated high antitumor efficacy and safety in order to develop more effective therapeutic strategies against melanoma. In vivo data demonstrated that DMPC-based Cuphen nanoliposomes significantly impaired melanoma progression with no toxic side effects [46].

A series of complexes of type [Cu(inh)_2_X_2_]·nH_2_O (inh is isoniazid; X is Cl (10), NCS (11) and NCO (12)) were incorporated in a nanostructured lipid system and exhibited an enhanced in vitro anti-*Mycobacterium tuberculosis* (Mtb) activity [47]. The same systems were considerably more selective than the unloaded compounds in interaction with *Escherichia coli* and *Staphylococcus aureus* strains; in some cases, the activity was up to 40 times higher [48].

The complex [Cu(ddc)_2_] (Hddc is diethyldithiocarbamic acid) (**13**) was tested, and proved more active than Hddc on glioblastoma (U251MG), human non-small cell lung cancer (A549) and breast cancer (MDA231-BR) cell lines. As a result, the compound was in situ synthesized into lipid vesicles and was developed as an injectable product for antitumor purposes [49]. The species [CuCl_2_]_n_ and [Cu(OH_2_)_4_]SO_4_∙H_2_O were embedded into stealth liposomes composed of 1,2-dipalmitoyl-sn-glycero-3-phosphatidylcholine (DPPC):cholesterol (CHOL):dipalmitoylphosphoethanolamine-N-[methoxy(polyethyleneglycol)-2000] ammonium salt (DPPE-PEG2000) and tested on human prostate cancer cell line (PC-3) both in vitro and in vivo. These formulations injected into tumor-bearing mice were effective in reducing the overall tumor burden [50]. Thermosensitive PEGylated liposomes based on DPPC and hydrogenated soybean phosphatidylcholine (HSPC) were used in order to incorporate the [Cu(ncr)_2_]SO_4_ (ncr is neocuproine or 2,9-dimethyl-1,10-phenanthroline) complex (**14**). These systems were effective in reducing the tumor growth of mouse colon carcinoma cells (C26) grafted in mice [51].

A liposomal formulation based on DSPC:CHOL containing a copper quercetin complex (**15**) with enhanced circulation longevity was recently developed and studied as system suitable for parenteral use in cancer treatment [52]. The vesicular assembly of metallosomes type developed by CHOL incorporation into cetylpyridinium chloride (CPC) surfactant in Cu(II) presence exhibits the ability to inhibit methicillin-resistant *S. aureus* (MRSA) strain [53]. 

The post-burn inflammation processes associated with thermally injured tissue gives rise to ROS overproduction. As a result, the influence of SOD1, free or liposomal encapsulated, on thermally injured tissue of rabbit skin was examined and the best result was achieved by spreading the liposomal formulation over the wounds. The liposomal formulation consisted in DPPC:CHOL/SA (stearylamine) in a molar ratio of 7:2:1 [54].

The therapeutic potential of some SOD1-containing polyethylene glycol (PEG)-liposomes was studied and compared with that of SOD1 entrapped in (SA)-liposomes and free SOD1 administered by (i.v.) injection in an arthritic rat model. Both small PEG-liposomes and SA-liposomes exhibited an enhanced therapeutic potential in comparison with SOD1, the PEG-liposomes being the most potent [55]. This enzyme was also covalently attached to the distal terminus of PEG polymer chains functionalized with maleimide, and this system was loaded at the surface of lipid vesicles composed of PC:CHOL in order to obtain SOD1-enzymosomes. The in vivo assays showed that SOD1-enzymosomes showed therapeutic activity in rat adjuvant arthritis both compared to liposomes with SOD1 encapsulated in their aqueous interior (SOD1-liposomes) and free SOD1. SOD1-enzymosomes, unlike SOD1-liposomes, have as therapeutic effect decreasing liver damage in a rat liver ischemia/reperfusion model. As a result, SOD1-enzymosomes were shown to be a new and successful therapeutic approach for oxidative stress-associated inflammatory diseases [56]. 

A highly efficient SOD1-loaded vesicle targeting the mucosal tissue was produced by enzyme encapsulation into liposomes composed of soybean lecithin (SLT), SA, phosphatidyl glycerol (PG) and CHOL with different charges. All types of liposomes were successfully coated with two types of low and medium molecular weight chitosans (CSs) that increased their mucoadhesive characteristics [57]. Superoxide dismutase attached to lecithin molecules was evaluated concerning the interaction with serum proteins and cells. The data indicated that the increased hydrophobicity of lecithinized SOD1 enhanced its association to both proteins and microdomains of plasma membrane. This formulation inhibited the SOD1 excretion, promoted its long-term retention in blood circulation and enhanced the cell internalization through endocytosis [58]. 

#### 2.1.2. Polymeric Systems

Several engineered systems were developed by back, embedding into a suitable biodegradable and biocompatible polymer such as PEG. The most studied bacc in PEGylation is SOD1. The SOD1-PEG conjugation was studied both in vitro and in vivo models in comparison with the free enzyme, and the following observations were drawn: (i) at the blood vessel level, this system provides a higher resistance to oxidative stress, improves endothelium relaxation and inhibits lipid peroxidation; (ii) in the heart, it is at least as effective as native SOD1 in both reperfusion-induced arrhythmias and myocardial ischemia treatment; (iii) in the lung, it reduces the oxygen toxicity and *Escherichia coli*-induced lung, and (iv) in kidney and liver ischemia, both enzyme forms were found to ameliorate reperfusion damage [59]. Having in view all these positive findings, it is surprising that this system has not yet started clinical trials.

The neuroprotective efficacy of SOD1 loaded in poly(d,l-lactide co-glycolide) (PLGA) nanoparticles (NPs), in cultured human neurons in oxidative stress induced by hydrogen peroxide, was also investigated. The results showed that this system is compatible with human neurons and its neuroprotective effect is dose-dependent and enhanced in comparison with both SOD1 and PEG-SOD1. The mechanism of SOD1-NPs efficacy appears to be their stability and the better neuronal uptake of enzyme after encapsulation [60]. The same system was tested in order to assay their efficacy in a rat focal cerebral ischemia-reperfusion injury model. SOD1-NPs were administered during the reperfusion via the intracarotid route to maximize their localization in the brain. The SOD1-NPs maintained blood brain barrier (BBB) integrity, thereby preventing edema, reducing the level of ROS and protecting the neurons from apoptosis [61].

SOD1 cross-linked to a cationic block copolymer, methoxy-poly(ethylene glycol)-block-poly(l-lysine hydrochloride) (PEG-pLL_50_) retains catalytic activity, has a narrow size distribution and a low cytotoxicity. This carrier has a prolonged ability to scavenge ROS production in cultured brain microvessel endothelial cells and central neurons. Moreover, in vivo it decreases the ischemia/reperfusion-induced tissue injury and improves sensorimotor functions in a rat middle cerebral artery occlusion (MCAO) model after a single i.v. injection [62]. This enzyme was also electrostatically attached to a synthetic poly(ethyleneimine)epoly(ethyleneglycol) (PEIePEG) polymer to form a polyion complex (SOD1 nanozyme). Intracarotid injection of this nanozyme into rabbits significantly inhibited the angiotensin II (AngII) intra-neuronal signaling [63].

In order to enhance in vitro viability and function of isolated neonatal pancreatic porcine cell clusters (NPCCs), SOD1 was co-cultured with PLGA microspheres (MS) that slowly releases the enzyme. This system significantly improved the morphology, viability and function of the incubated NPCCs, and the slow SOD1 release could contribute to overcome the scarcity of transplant in patients with type 1 diabetes mellitus (T1DM) [64]. A modified method was used for both PLGA and poly(d,l-lactide) (PLA) MS preparation together with mannitol, trehalose, and PEG400 co-encapsulation for SOD1 stabilization. In vitro activity retention was evaluated by both nicotinammide adenine dinucleotide (NAD) oxidation and H_2_O_2_ consumption assays. SOD1 encapsulation efficiency resulted in 30 to 34% for PLA MS and up to 51% for PLGA MS. SOD1 in vitro activity was preserved in all systems and was better stabilized in PLGA MS [65].

Another efficient system was developed by SOD1 encapsulating in the aqueous space of some porous polymersomes obtained by coassembly of poly(ethylene glycol)-polybutadiene (PEG-PBD) and poly(ethylene glycol)-*block*-poly(propylene oxide)-*block*-poly(ethylene glycol) (PEG-PPO-PEG). In vivo studies in the rats evidenced the polymersomes ability to prevent neuropathic pain associated with nerve root compression more effectively in comparison with free enzyme [66].

Two cationic block copolymers poly(ethylene glycol)-*b*-poly(l-lysine) (PEG-PLL) and PEG-*b*-poly(aspartate diethyltriamine) (PEG-DET) were engineered for chronic dosage of SOD1. These systems are well-tolerated by brain microvessel endothelial/neuronal cells and reduce the infarct volumes in a mouse model of ischemic stroke [67,68,69]. 

Besides SOD1, a series of other bacc species were embedded into polymeric matrix. Among these, nanogel composed of *N*-[3-(dimethylamino)-propyl]methacrylamide (NDPMA) and sodium copper chlorophyllin (SCC) was synthesized and studied as a nanocarrier for photothermal therapy. The positively charged NDPMA at tertiary amine groups could facilitate the nanogel access into the cells through electrostatic attractions. The photothermal effect of SCC-containing nanogel via green laser exposure generated a mortality enhancement by 60% in the murine fibroblast (L929) cell line [70]. The SCC was also incorporated into a thermo-responsive polymeric nanogels formed by *N*-isopropylacrylamide and *N*-(hydroxymethyl)acrylamide copolymerization. The combination of hyperthermia and temperature-induced drug release via green laser irradiation greatly enhanced cell mortality of L929 cells to a maximal extent [71], and as a result, such a system could have a great potential in antitumor therapy. 

The complex [Cu(dttct)] (**16**) (dttct is dibenzo[e,k]-2,3,8,9-tetraphenyl-1,4,7,10-tetraaza-cyclododeca-1,3,7,9-tetraene) embedded within a poly(vinyl chloride) (PVC) was used for catalytic generation of nitric oxide in the presence of nitrite and ascorbic acid as reducing agent. This system showed an effectively control in both the formation and dispersion of nitrifying bacteria biofilm [72]. This complex immobilized together with [Fe(dttct)] onto the same matrix composed of a system able to generate nitric oxide (NO) from endogenous nitrite through Cu(II) conversion to active Cu(I). Besides nitrifying bacteria biofilm dispersion, the antibiofilm activity of these material was evidenced against *Bacillus* species [73].

The slow release of Cu(II) from [Cu_3_(btc)_2_] (**17**) (H_3_btc: benzene tricarboxylic acid) NPs embedded in an antioxidant poly-(polyethyleneglycol citrate-*co*-*N*-isopropylacrylamide) (PPCN) hydrogel was set up. Such a system reduced the Cu(II) toxicity and enhanced the wound healing process for diabetic mice by promoting a plethora of processes such as cell migration, angiogenesis and collagen deposition [74]. This compound, embedded in PVC, reduced the bacterial attachment of both *S. aureus* and *E. coli* strains [75].

Polyaniline nanoshuttles coated with polyethylene glycol (PEG) and codoped with both Cu(II) and vincristine were developed as multifunctional, positive charged and long-circulating systems. These characteristics explain their higher accumulation in human oral epithelial carcinoma vincristine-resistant tumors (KBV) that further improved the efficacy of tumor theranostics, such as those using chemotherapy and photothermal therapy [76]. The [Cu(ddc)_2_] (**13**) moiety coordinated to carboxylic groups from poly(ethylene glycol)-*b*-poly(ester-carbonate) generated a highly stable system able to inhibit A549 cells proliferation both in vitro and in vivo [77]. The interest for this bacc is generated by the fact that dsf exhibits an excellent in vitro antiproliferative activity mediated by (**13**) as its metabolite.

The Cu(II) complex with poly[2-(*N*-oxide-*N*,*N*-dimethylamino)ethyl methacrylate] (OPDMA) is a water-soluble zwitterion that can deliver metallic ions into tumor tissues, and was developed in order to potentiate the dsf antitumor efficacy [78]. A delivery system of [Cu(dpphen)(imc)_2_] (**18**) (dpphen is 4,7-diphenyl-1,10-phenanthroline, Himc is indomethacin) with methoxy poly(ethylene glycol)-*b*-poly(d,l-lactic-co-glycolic) acid (PEGPLGA) copolymer exhibits low toxicity and ability to selectively kill breast cancer stem cells over bulk ones. The cytotoxic effect comes from both intracellular ROS generation and cyclooxgenase-2 (COX2) inhibition [79].

A series of water soluble polymer copper(II) complexes (pcc) were obtained by some species with aromatic amine ligand coordination to the nitrogen atoms from branched polyethyleneimine (BPEI) and tested either for antitumor or for antimicrobial activity. Polyethyleneimine (PEI) is a cationic polymer studied as an alternative to liposomal routes of gene delivery [80]. This polymer acts as chelating agent, has good water solubility, a high number of functional groups and stability [81,82,83,84,85].

The systems [Cu(phen)(L-Thr)(BPEI)]ClO_4_∙2H_2_O (**19**) [81] and [Cu(phen)(L-phe)(BPEI)]ClO_4_∙4H_2_O (L-Thr: L-threonine, L-phe is L-phenylalanine) (**20**) [82], with various degree of copper(II) coordinated in the polymer matrix, were prepared and characterized. Studies concerning their binding abilities to calf thymus DNA (CT DNA) has been investigated, and the results indicated that the copper(II) content in the polymer backbone have a marked effect on the binding affinity. These pcc showed a good antimicrobial activity against some bacteria and fungi, comparable with that of standard drugs ciprofloxacin and clotrimazole, respectively. On the other hand, the systems [Cu(phen)_2_(BPEI)]Cl_2_·4H_2_O (**21**) [83] and [Cu(bpy)_2_(BPEI)]Cl_2_∙4H_2_O (bpy is 2,2′-bipyridine) (**22**) [84,85], besides good antimicrobial activity, exhibited cytotoxicity against human lung cancer cells (NCI-H460). Cytological changes, such as chromatin fragmentation, binucleation, cytoplasmic vacuolation, nuclear swelling, cytoplasmic blebbing as well as late apoptosis and/or necrosis, were also observed during this experiments. The presence of multiple copper(II) complex moieties and free NH groups, in a single polymer molecule, enhanced the binding ability of these species to DNA through electrostatic, van der Waals and hydrogen bonding and/or intercalation [81,82,83,84,85].

The chemical resistance, low cost and versatile properties of polypropylene (PP) make it an attractive polymer for medical applications. A PP film functionalized with glycidyl methacrylate as a spacer arm and iminodiacetate moiety as a chelating group for Cu(II) has an anti-adhesive effect in case of *E. coli* [86]. A series of 2,6-pyridinedicarboxylate-based polyesters employing several diols with different aliphatic chain were synthesized and complexed with copper(II). The composites were tested for their antibacterial potential and were found to effectively resist *P. aeruginosa* attachment and colonization [87]. Additionally, the Cu(OH_2_)_4_ moiety was used to link together the systems obtained by castor oil phosphorylation. The generated polymer exhibits a good antibacterial activity against *B*. *cereus* and *E. coli* strains [88].

Chlorophyllin was encapsulated together with CdSe/ZnS as quantum dots into PLGA. This coformulation allows the generation of ROS upon excitation both in an aqueous medium and in cells, thus exhibiting potential in photodynamic therapy [89]. Some natural polymeric materials, such as polysaccharides (chitosan, hyaluronic acid, sodium alginate, cellulose) and proteins (sponge skeleton, albumin), have been used in recent years for bacc incorporation. Their useful features, which attracted attention, include low cost, biodegradability, renewability and high compatibility with human body [5].

Chitosan (CS) is a polyglucosamine with chelating abilities, as well as antimicrobial and hemostatic activity and cytocompatibility and mucoadhesion ability [90]. By CS mixing with chlorophyllin, an insoluble salt-like material was obtained. This system has the ability to efficiently trap polycyclic compounds such as heterocyclic amines and aflatoxin B_1_, thus suppressing their mutagenic and carcinogenic behavior [91]. The SOD1 encapsulated CS microspheres were prepared by changing both pH and PEG addition, and it was observed that the protein activity remained within acceptable limits after the release of these formulations [92].

A water soluble complex of Cu(II) with chitosan have the ability to interact with DNA in two steps, consisting of electrostatic interaction and intercalation into the base pair [93]. On the other hand, chitosan was derivatized by condensation with salicylic aldehyde, and Cu(II) was coordinated into the Schiff base site. The nanoparticles of this system interact also with DNA via electrostatic interaction, while the aromatic moieties are able to intercalate. Moreover, this macromolecular complex inhibited the liver cancer (SMMC-7721) cell line, its activity being enhanced in comparison with both CS and free CS-Schiff-base [94].

Some Cu(OH_2_)X (**23**) (X is CH_3_COO) and (**24**) (X is ClO_4_) moieties with Schiff base formed by CS derivatization with 2-hydroxy-3-metoxybenzaldehyde were tested in vitro against chronic myelogenous leukemia (K562) and osteosarcoma cancer (MG-63) cell lines. The good activity observed was related to an apoptotic effect [95]. On the other hand, Casiopein III-ia (**8**) loaded into CS nanoparticles increases the life span in mice transplanted with melanoma (B16) tumor cells [96]. The fibrous scaffolds obtained by homogeneous blend of copper(II)-CS complex and polycaprolactone generates a significant increase in the secretion of vascular endothelial growth factor (VEGF), a factor that can have high impact on endothelial cells, guiding angiogenesis both ex vivo and in vivo [97].

An antibacterial film composed of CS and [Cu_3_(btc)_2_] (**17**) species has the ability to slow release copper ions, reduced cytotoxicity and antibacterial activity against *E. coli* and *S. aureus* strains. Furthermore, in vivo results on mice wounds infected with *S. aureus* revealed that this film could simultaneously kill bacteria and promote vessel regeneration, resulting in an enhanced wound closure rate during the local infection therapy process [98]. Some copper(II)-CS complexes were fabricated via in situ precipitation and tested on mouse embryonic fibroblasts and both *S. carnosus* and *E. coli* strains. Combined cells analysis and bacterial studies identified a threshold concentration at which the material shows antibacterial properties without significantly affecting the fibroblast viability **[99]**. Copper(II) complexes with CS functionalized with a Schiff base derived from amino benzoic acid and acetyl acetone was developed and tested on *Phytophthora capsici* Leonian. The results suggest that the system causes mitochondrial injury with enhanced ROS and reduced ATP levels, thereby killing this fungus [100].

Hyaluronic acid (HA), a linear polysaccharide found in physiological fluids, is another species able to coordinate copper(II) and promote an angiogenic response through endothelial cells mobilization [101]. A hydrogel based on HA with Cu(II) coordinated through the amidic and carboxylate oxygen atoms, evaluated in vitro by using fibroblast (3T3) cells and in vivo on rats, showed proangiogenic activity by stimulating the growth of new capillary vessels without any inflammatory reaction [102].

The complex [Cu(ppt)]_n_ (H_2_ppt is polymerized-*p*-phenylenediamine-5,10,15,20-tetra-(4-aminophenyl)porphyrin) embedded in HA generated a multifunctional nanoplatform with a good antitumor activity potentiated by a synergistic photo-/chemodynamic and immunotherapy. This system exhibits antimetastatic activity, ability to suppress the tumor growth as well as immune response activation in colon carcinoma (CT26) tumor cells both in vitro and mice grafted or injected ones. It was observed that the light excitation enhanced the therapeutic effect that was related to ROS generation through Fenton-like reactions [103].

An amylopectin modified with diethylenetriaminepentaacetic acid residues for copper chelation was prepared as hydrogel. The product conjugated with basic fibroblast growth factor (bFGF) has the ability to regulates its controlled release due to hydrogel biodegradation, thus resulting in a prolonged neovascularization in mice. The Cu(II) decreased the rate of bFGF release, thus showing antitumor potential [104]. The same team developed a Cu(II) complex with dextran modified with diethylenetriaminepentaacetic acid that enabled tumor necrosis factor α (TNF) to coordinative conjugate to this system. Its i.v. administrations suppressed tumor growth in tumor-bearing mice by a higher TNF tumor accumulation [105].

Sodium alginate (SAG), a linear polysaccharide present in the cell walls of various seaweeds, has along the backbone several hydroxyl and carboxyl groups that can be functionalized. This species was successfully used as oral delivery systems [106]. The micelles composed of Cu(II) complex with this polysaccharide derivatized with *N*-(2-hydroxyethyl)-*N*,*N*-dimethyldodecan-1-aminium bromide cationic surfactant was evaluated against several bacteria (*B. subtilis*, *S. aureus*, *E. coli* and *Pseudomonas aeruginosa*) and fungi (*C. albicans* and *Asperigllus niger*). The system exhibits an enhanced antimicrobial activity on all tested strains in comparison with derivatized SAG and its Co(II) complex but lower in comparison with the Zn(II) species [107].

Cyclodextrins (CD), as cyclic polysaccharide containing D-glucopyranose units linked with α-1,4-glucosidic bonds, are classified as α, β or γ depending on the number of such units from their structure [108] Their ability to be functionalized and to retain complexes into their cavity was exploited in order to develop useful carriers. A hydrogel film composed of carboxymethyl cellulose (CMC), cellulose nanocrystals (CNC) and hydroxypropyl β cyclodextrin (HP-β-CD) cross-linked by citric acid was prepared and studied for the controlled release of the neohesperidin-copper(II) complex. The system exhibits cell compatibility and a low cytotoxicity, and as a result can be studied as a bacc delivery material with controlled release [109]. The soluble complex [Cu(chz)Cl] (Hchzis 6-hydroxychromone-3-carbaldehyde-(3′-hydroxy)benzoylhydrazone) was spontaneously included in heptakis-2,6-*O*-dimethyl-β-cyclodextrin (DMβCD) and the inclusion complex was proven to bind CT DNA in an intercalative mode [110].

Another biodegradable and biocompatible polysaccharide, cellulose acetate (CA) was functionalized with [CuL_2_Cl_2_] (L is 2-fluoropyridine) in order to obtain microfibers via the electrospinning technique. The obtained system was found to have considerable antibacterial effect against *E. coli* and MRSA [111]. Copper(II) was also coordinated to a cellulose-based material functionalized at aldehyde groups by condensation with glycine. This system enhanced the antibacterial properties of the cellulose fibers against *S. aureus* and *E. coli* strains [112]. The material, consisting of *Hippospongia communis* sponge skeleton and SCC, reduced the growth of *S. aureus* in dependence with the SCC content. It was also observed that the chlorophyllin adsorption on *H. communis* sponge skeletons varies as a function of the pH and ionic strength [113].

An SOD1 carrier was obtained by enzyme conjugation with both mouse vascular cell adhesion molecule 1 (VCAM-1)-targeted nanobodies and an albumin-binding arm (VCAM/ALB8). The bispecific system VCAM/ALB8 was superior in comparison with VCAMelid in enhancing both the circulation time and organ targeting of the enzyme [114]. Other systems were developed by SOD1 conjugation with ferritin [115], with antibodies to plasmalemmal vesicle-associated protein (Plvap) [116] or to platelet endothelial cell adhesion molecule (PECAM) [117] for a targeted delivery of enzymatic cargo to endothelial caveolae-derived endosomes.

The nanosystem formed by bovine serum albumin (BSA), Cu(II) and 5-nitro-8-hydroxyquinoline, a known anticancer agent, was developed as a selective tumor targeting formulation. Its cytotoxic effect was enhanced in comparison with components in mouse breast (4T1), human non-small cell lung (A549) and human cervical (HeLa) cancer cells and moreover without a systemic toxicity [118].

#### 2.1.3. Dendrimers

Dendrimers are highly branched three-dimensional macromolecules with a globular and monodisperse structure and groups able to coordinate metallic ions, properties that afford their applications as Cu(II) carriers for antitumor and antimicrobial systems, including for biofilm formation prevention. A proper selection of dendritic scaffold, generation type and ligand attached to Cu(II) can lead to a potent activity that can overcome the limitations of traditional therapies [119]. The main type used for Cu(II) delivery are nitrogen-, phosphorus- and silicone-based dendrimers.

A second-generation poly(propylene imine) dendrimer modified with acridine and loaded with Cu(II) showed a good antimicrobial activity on *B. cereus* and *C. lipolytica* and low cytotoxicity against the HEp-2 cell line. Moreover, modified cotton fabrics with this dendrimer prevent biofilm formation in case of *B. cereus* and *P. aeruginosa* strains but do not exhibit cytotoxicity against the Human epithelial type 2 (HEp-2) cell line [120].

A first generation polyamidoamine (PAMAM) dendrimer functionalized with 1,8-naphthalimide units was synthesized, loaded with different amount of Cu(II) (**25**) (Figure 2) and attached on the cotton surface. Both materials had the ability to reduce bacterial growth and prevent the biofilm formation in case of *B. subtilis*, *B. cereus* and *Acinetobacter johnsonii*, the best effect being observed on the last strain [121]. On the other hand, the second generation PAMAM dendrimer modified with 4-(*N,N*-dimethylaminoethyloxy)-1,8-naphthalimide and conjugated with *cis*-Cu(NO_3_)_2_ moiety (**26**) (Figure 2) was deposited on cotton fabric, and this material was found to be active against biofilm produced by *B. cereus*, *P. aeruginosa* and *C. lipolytica* strains [122].

Among the phosphorus-containing dendrimers, those bearing iminopyridine (IP) terminal groups have good ability to coordinate Cu(II) [123]. As a result, some phosphorus multidentate dendrimers of generation Gn (*n* = 1 to 3) decorated with IP and coordinated to CuCl_2_ moiety were developed and tested for antiproliferative activity against a panel of tumor cell lines. The complex with a system derived from *N*-(pyridin-2-ylmethylene)ethanamine exhibits a very potent antiproliferative activity (>80%) against epidermal carcinoma (KB) and leukemia (HL60) cell lines [124]. The mechanism of action consists of the proapoptotic protein Bax activation pathway [125]. The EPR studies evidenced the high stability of G3 dendrimer functionalized with *N*-(di(pyridine-2-yl)methylene) ethanamine moiety in correlation with its good activity [126].

A third generation of phosphorus dendrimer decorated with (IP) groups as Cu(II) complex was developed as an agent with enhanced tumor accumulation. This system inhibits the pancreatic cancer (SW1990) cell line through the apoptosis activation via up- or downregulation of some genes [127]. Metallophosphorus dendrimer (G3, 48 terminal groups) with IP units was prepared with both Au(III) and Cu(II), and a synergistic effects of the two cations on the antimicrobial activity was observed in case of ATCC standard strains *S. aureus*, *E. coli*, *P. aeruginosa* and *C. albicans* as well as against drug-resistant clinical isolates *S. aureus*, *Enterococcus faecalis* and *C. glabrata*. The minimal inhibitory concentration (MIC) values fall in a concentration range of 3.5–500 mg/L and the most sensitive strain was *E. coli* [128].

A G1 carbosilane dendrimer decorated with IP groups and conjugated with Cu(OH_2_)(ONO_2_)_2_ moiety was developed as water-soluble, stable and potent bacteriostatic and bactericide species against planktonic *S. aureus* and *E. coli* strains. Furthermore, it prevents the formation of *S. aureus* biofilm at a low concentration and is not hemotoxic [119]. The corresponding system with Cu(OH_2_)Cl_2_ moiety exhibits a moderate cytotoxicity in myeloid (U937) cancer cells. The system induces cell death through the mitochondria–lysosome system as well as vacuole formation through autophagy [129]. A comparative study of such G1 and G2 dendrimers conjugated with Cu(OH_2_)X_2_ (**27**) (X: is ONO_2_) and (**28**) (X is Cl) moieties (Figure 2) revealed that the increase in generation and changing nitrate with chloride, produced an increased in vitro cytotoxic effect in cervix (HeLa), normal and resistant breast cancer (MCF7 and HCC1806), advanced prostate cancer (PC3) and colorectal tumor (HT29) cell lines. It is worth to mention the good activity against the resistant tumor cells with IC_50_ values below 3.4 µM in PC3 and below 1.9 µM in HCC1806. Moreover, one of the nitrate derivatives both inhibited proliferation and decreased the adhesion to collagen type-I PC3 cells in an experimental ex vivo mice model of human prostate cancer [130]. Similar systems from G0 to G2 modified with Cu(OH_2_)(NO_3_)_2_ or Cu(OH_2_)Cl_2_ moieties were also studied concerning their cytotoxic properties toward healthy peripheral blood mononuclear cells (PBMC) and leukemia (1301 and HL-60) cancer cell lines. The cytotoxicity assays showed that all compounds were more active on 1301 in comparison with HL-60 cell line with a low toxicity on PBMC [131].

Carbosilane G1 copper (II) metallodendrimers modified with Schiff base and loaded with small interfering RNA (siRNA) were developed as an antitumor strategy. Cell viability assays performed by flow cytometry indicated interactions between these dendrimers and pro-apoptotic siRNAs genes in MCF-7 [132].

### 2.2. Inorganic Based Carriers

The bacc [Cu_3_(btc)_2_] (**17**), embedded in titanium oxide film and 316L stainless steel as materials with a good biocompatibility with both endothelial and macrophages cells, inhibited the bacterial adhesion of both *S. aureus* and *E. coli* strains [75]. The same species immobilized on titanium foils exhibited ideal NO release and a synergistic role of NO and Cu(II) in inhibiting platelet activation, suppressing macrophages adhesion and proliferation in order to reduce hyperplasia. The ex vivo and in vivo experiments indicated an enhancement of re-endothelialization and anti-hyperplasia by this immobilized coating with the potential for cardiovascular applications [133].

CSS was also embedded in hydrotalcite [134] or graphene oxide [135] in order to obtain materials with bactericidal effect. The hydrotalcite based system is active on *E. coli*, *Enterobacter aerogenes*, *Salmonella enterica* and *S. aureus* strains and exhibits a low toxicity [134]. In case of graphene oxide based nanomaterials functionalized with chlorophyllin and chlorophyllin-Zn(II), the last one exhibited the highest antibacterial activity against *Escherichia coli*. Further investigations show that the bacteria incubation with these nanomaterials caused cell death through cellular integrity loss [135].

SOD1 was covalently immobilized onto oxidized multiwalled carbon nanotubes (MwCNTs) by using a diimide activated amidation reaction. The obtained material produced a low level of ROS and an increased enzyme activity into human keratinocytes (HaCat) cells [136]. Mesoporous vaterite (CaCO_3_) was also used as a vector for SOD1 delivery for an ophthalmological formulation [137].

The [CuL_2_] (HL is acetic acid, acetyl acetone) encapsulation into functionalized nanostructured titanium dioxide (TiO_2_) provided porous nanoparticles with a promising in vitro antitumor activity and no significant toxic effect on rat glioma (C6 and RG2), human glioma (U373) and mouse melanoma (B16) cell lines [138].

Mesoporous silica (MSS) were also studied as inorganic drug-delivery systems due to their attractive properties, such as a high surface area, tunable pore/particle size/morphology, a very high chemical stability as well as an easy surface functionalization [139]. As a result, some examples of bacc embedded in MSS matrix have been reported. One of them was obtained by diamine functionalized MS combination with curcumin, Cu(II) and immobilized AgNPs, and showed an excellent photodynamic inactivation of antibiotic-resistant *E. coli.* The activity was assigned by a synergistic effect of Cu(II), silver and curcumin [140]. MSS functionalized stepwise with 3-aminopropyltriethoxysilane, 2-hydroxy-3-methoxybenzaldehyde and Cu(II) showed an inhibitory effect and growth retardation on *S. aureus* and bacteriostatic effect against *E. coli* [141]. MSS NPs containing a maleamato ligand was used to coordinate copper(II) ions and evidenced an improved activity against the same strain [142].

The SBA-15 (Santa Barbara Amorphous material 15) MSS grafted with triethoxysilylpropylmaleamic acid (maleamic) and triethoxy-3-(2-imidazolin-1-yl)propylsilane (imidazoline) was loaded with copper(II) in the presence of 5,5′-dimethyl-2,2′-bipyridine as an ancillary ligand. Although the material with maleamic acid contains a lower amount of Cu(II), it exhibits a higher activity against *S. aureus* and *E. coli* strains, properties related to ROS production and with the ability to interfere in both peptidoglycan synthesis and glutathione metabolism [143].

The SOD1 mimics, [CuZn(dien)_2_(μ-Im)(ClO_4_)_2_]ClO_4_ and [Cu_2_(dien)_2_(μ-Im)(ClO_4_)_2_]ClO_4_ (Him is imidazolate, dien is diethylenetriamine), were also encapsulated in MSS, and the obtained materials showed an improved anti-inflammatory activity [144].

An efficient targeted delivery platform based on folate-receptor was developed by [Cu(L)(dppz)]^+^ (LH is 2-[(2-dimethylaminoethylimino)methyl]phenol, coordinated ligand, dppz = dipyrido[3,2-a:2′,3′-c] phenazine) encapsulation into MSS. This system exhibits selective accumulation and high cytotoxicity on breast cancer cells MCF-7 and MDA-MB-231 [145].

### 2.3. Hybrid-Based Materials

Among hybrid materials, some Ca-alginate (AG) systems with a porous carbonate cores were loaded with SOD1, and their efficacy in protection of the enzyme activity in simulated intestinal fluid with trypsin was demonstrated [146].

The gold nanoparticles modified with amine terminated PEG and decorated with biotin were studied as a delivery system for the [CuL] (H_2_L is diacetyl-bis(*N*4-methylthiosemicarbazone)) complex with specificity for cancer cells. The Cu(II) complex was attached to a linker containing disulfide bond loaded onto the gold nanoparticle surface. Both in vitro and in vivo studies proved the activity on HeLa cells, both free and xenograft in mice, with a 3.8-fold reduction in tumor volume during the treatment [147]. Copper(II) coordinated at glucosamine and *N*-acetylglucosamine groups of CS generated some coatings fabricated by electrophoretic deposition on a 316L stainless substrate. The test performed on *S. aureus* and *E. coli* evidenced the bacterial growth inhibition in the early stages of implantation without any cytotoxicity for all systems containing different amount of Cu(II) [148].

Among the magnetic nanoparticles, magnetite (Fe_3_O_4_) has unique properties, such as low toxicity, superparamagnetism, high surface area, biocompatibility and a chemically modifiable surface. An organic–inorganic hybrid material based on magnetic mesoporous silica Fe_3_O_4_@MCM-41 was functionalized with *N*-(2-aminoethyl)-3-aminopropyltrimethoxysilane and grafted with a Schiff base for Cu(II) coordination. This material conjugated with streptomycin completely inhibited the growth of both *S. aureus* and *E. coli*, the effect being more prominent in the presence of a magnetic field [149].

## 3. Conclusions

Nowadays, the medical interest for copper complexes arises from their used as therapeutic agents in Menkes disease, inflammatory conditions, iron deficiency anemia or various skin conditions, or as drug candidates investigated in clinical trials for the efficacy in the treatment of amyotrophic lateral sclerosis or cancer. The small number of copper-based drugs that are in current use is a result of a reduced stability both in water and acidic environment and of an unfavorable balance between water solubility and lipophilicity. In order to overcome these problems, a series of biological active copper complexes were combined with a wide range of organic, inorganic or hybrid (organic-inorganic)-based carriers. The most studies were performed with known drugs, such as SOD1 and chlorophyllin, that were embedded both in organic and inorganic matrices. The association with both kinds of matrices have improved the pharmacological profile of drug; the most promising approach seems to be the liposomal embedding. Generally, the studies were performed for anti-inflammatory, antimicrobial and antitumor species. Important is that many formulations were either on biofilm embedded microorganisms or on resistant bacteria or tumor cells.

## Figures and Tables

**Figure 1 molecules-25-05830-f001:**
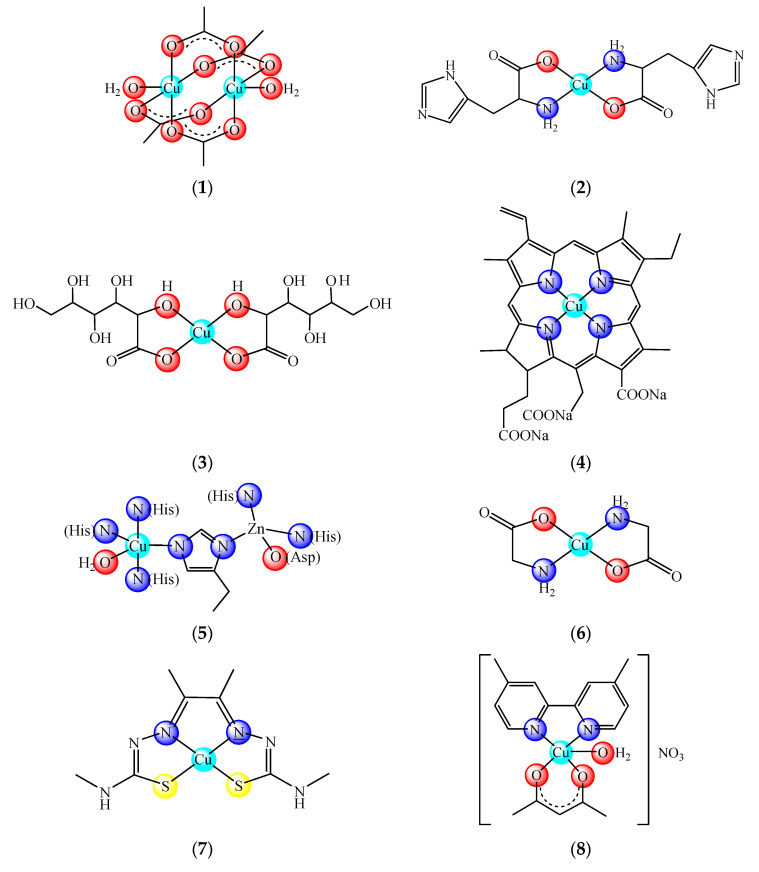
Copper-based drugs: [Cu_2_(CH_3_COO)_4_(OH_2_)_2_] (**1**) and [Cu(His)_2_] (**2**) for relieving symptoms in Menkes disease, [Cu(glu)_2_] (**3**) for iron deficiency anemia, chlorophyllin (**4**) for wounds odors elimination, radiation burns, inflammatory diseases and liver cancer prevention, SOD1 (**5**) for inflammatory diseases, diabetic complications, atherosclerosis, Alzheimer’s disease, cancer and rheumatic arthritis, [Cu(Gly)] (**6**) for skin conditions, [Cu(atsm)] (**7**) for amyotrophic lateral sclerosis and [Cu(dmbpy)(acac)(OH_2_)]NO_3_ (**8**) for cancer.

**Figure 2 molecules-25-05830-f002:**
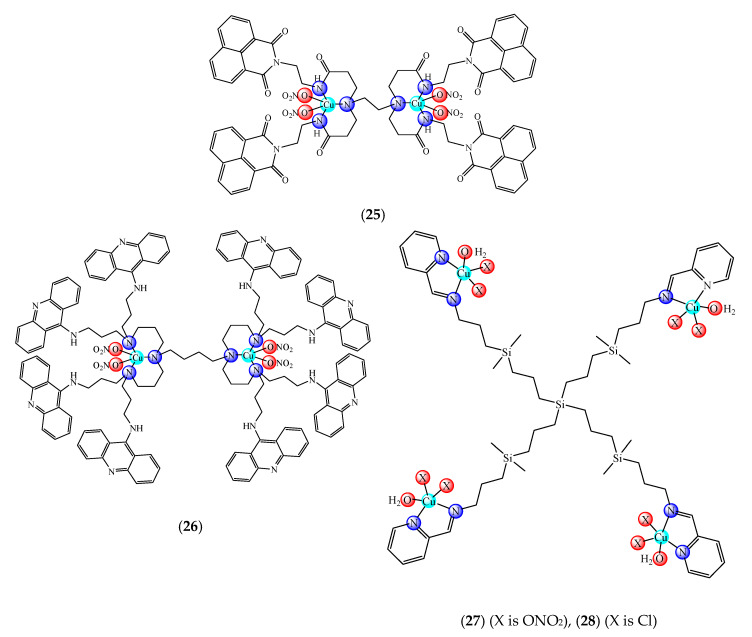
Bioactive dendrimeric complexes (**25**)–(**28**).

**Table 1 molecules-25-05830-t001:** Examples of biologically active Cu(II) complexes embedded in organic matrix.

Complex/Conjugated Moiety	Matrix	Activity	Other Tests	Ref
**Liposomal Formulations**				
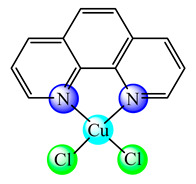 (**9**)	ePC ePC:DMPC:DPPC: DSPE-PEG DMPC:Chol:DSPE-PEG DMPC:CHEMS: DSPE-PEG2000	A431 (IC_50_: 10/8.3 μM), HaCat (IC_50_: 5.3/5.8 μM), MNT-1 ((IC_50_: 4.8/4.4 μM)), B16F10 (IC_50_: 4.5/5.1 μM), C26 (IC_50_: 5.8/4.4 μM) male BALB/c mice B16F10 (IC_50_: 5.1/2.1/2.7 μM)	In vitro hemolysis, hepatotoxicity Hepatic biochemical parameters, caspase activity	[45,46]
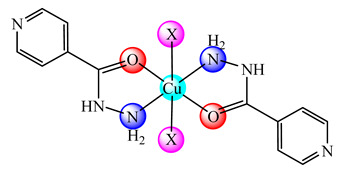 X is Cl (**10**), NCS (**11**), NCO (**12**)	CHLN:PC	*M. tuberculosis* (MIC: 0.397, 0.219, 0.313 μg/mL) *S. aureus* ATCC 25,923 (MIC: 250, 500, 125 μg/mL), *E. coli* ATCC 25,922 (MIC: 125, 125, 500 μg/mL)	Vero cell line cytotoxicity ((IC_50_ 109.5–319.3 μg/mL), macrophage, *Artemia salina* (brine shrimp)	[47,48]
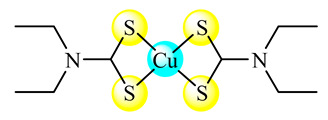 (**13**)	DSPC:CHOL: DSPE-PEG2000	Female CD-1 mice		[49]
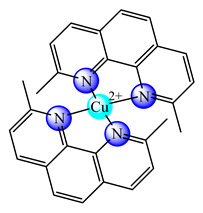 (**14**)	DPPC:HSPC	HT-29 (IC_50_: 0.2–10.1 μM) C26 (IC_50_: 0.2–4.2 μM) C26 grafted to BALB/c mice		[51]
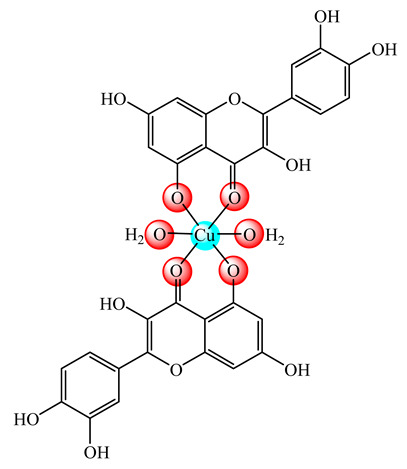 (**15**)	DSPC:CHOL (55:45 molar ratio)	parenteral administration		[52]
**Polymer formulations**				
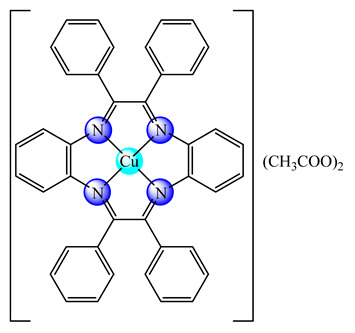 (**16**)	PVC	Biofilm produced by nitrifying bacteria inhibition		[72,73]
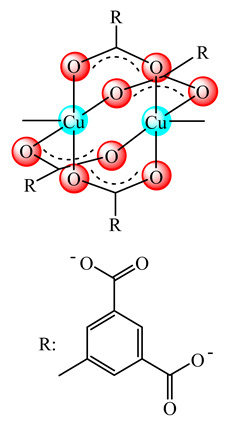 Basic moiety in (**17**)	PPCN PVC	Wound healing in diabetic mice Biofilm of *S. aureus*, *E. coli* inhibition	Cell migration, angiogenesis and collagen deposition	[74,75]
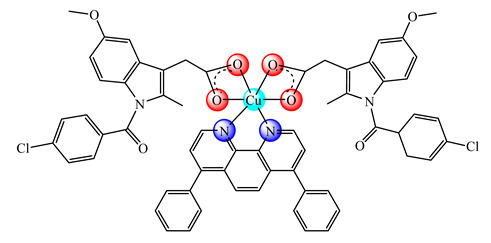 (**18**)	PEGPLGA	HMLER (IC_50_: 7.38 μM) HMLERshEcad (IC_50_: 2.21 μM)	ROS production COX-2 inhibition	[79]
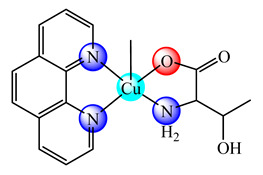 (**19**)	BPEI	*S. aureus*, *B. subtilis*, *E. coli*, *P. aeruginosa*, *C. albicans* (diameter of inhibition zone)	CT DNA interaction	[81]
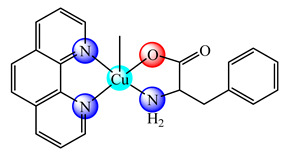 (**20**)	BPEI	*S. aureus, B. subtilis, E. coli, P. aeruginosa, C. albicans* (diameter of inhibition zone)	CT DNA interaction	[82]
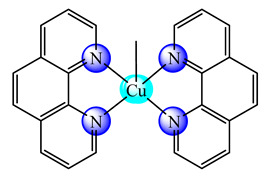 (**21**)	BPEI	*S. aureus, B. subtilis, E. coli, P. aeruginosa, C. albicans* (diameter of inhibition zone) NCI-H460 cells (IC_50_: 13.2 μg/mL)	CT DNA interaction	[83]
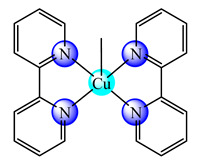 (**22**)	BPEI	*S. aureus, B. subtilis, E. coli, P. aeruginosa, C. albicans* (diameter of inhibition zone) NCI-H460 (IC_50_: 90–95 μg/mL)	CT DNA interaction	[84,85]
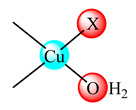 (**23**) X: CH_3_COO, (**24**) ClO_4_	Chitosan Schiff base with 2-hydroxy-3-metoxybenzaldehyde	K562 (IC_50_: 1 μg/mL) MG-63 (IC_50_: 25 μg/mL)	Apoptosis studies	[95]

Branched polyethyleneimine (BPEI), cholesteryl hemisuccinate (CHEMS), choline (CHLN), cholesterol (CHOL), dimyristoyl phosphatidylcholine (DMPC), 1,2-dipalmitoyl-sn-glycero-3-phosphatidylcholine (DPPC), dipalmitoylphosphoethanolamine-N-[methoxy(polyethyleneglycol)-2000] ammonium salt (DPPE-PEG2000), 1,2-distearoyl-sn-glycero-3-phosphocholine (DSPC), distearoyl phosphatidylethanolamine (DSPE), hydrogenated soybean phosphatidylcholine (HSPC), phosphatidylcholine (PC), poly(ethylene glycol (PEG), methoxy poly(ethylene glycol)-*b*-poly(d,l-lactic-co-glycolic) acid (PEGPLGA), poly-(polyethyleneglycol citrate-*co*-*N*-isopropylacrylamide) (PPCN), polyvinyl chloride (PVC). Chronic myelogenous leukemia (K562), human colorectal adenocarcinoma (HT-29), human epidermoid carcinoma (A431), human keratinocytes (HaCaT), human melanotic neuroectodermal tumor (MNT-1), osteosarcoma (MG-63), mammary epithelial cell lines (HMLER, HMLERshEcad), murine colon cancer cells (C26), murine melanoma (B16F10), U251MG (glioblastoma cell line).

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
