# Peer review of "Improvement in the Pharmacological Profile of Copper Biological Active Complexes by Their Incorporation into Organic or Inorganic Matrix"

_molecules, 2020, doi:10.3390/molecules25245830_

Round 1
Reviewer 1 Report
I read this review with interest and found a lot of interesting things. It was interesting to find ways to incorporate biologically active copper complexes into various types of matrix. I think this review will find its readers.
I recommend it for publishing after a minor revision.
The authors should first complete the text with a summary of the toxicity of copper.
I recommend the review of Osredkar and Sustar as an additional reference [10.4172/2161-0495.S3-001].
The table 1 should be corrected. 1. Add information about matrix for complex 18 in Polymer formulations. I don’t understand the nature of monodentate ligands in complexes 19-24. Iodid, CH3- or H2O? Please clarify.
The text contains some authors corrections.
I recommend to read the text carefully, I found some grammatical mistakes: line 39 -
tretment; line 59 - cupro-proteins; line 61 - cupro-enzymes; line 75 – cooper; line 440 – atached.
Please change the ligand decoding format: “phen is 1,10-phenantroline” instead of “phen: 1,10-phenantroline”.
Line 261: [Fe(dttct)] instead of [Fedttct].
I don’t understand the word “bacc”.
Reviewer 2 Report
The manuscript of the review entitled "Improvement in the Pharmacological Profile of Copper Biological Active Complexes by their Incorporation into Organic or Inorganic Matrix" authors Mihalea Badea, Valentina Uivarosi and Rodica Olar is generally well presented. The theme concerning of biological activities of Cu-complexes and their conjugates is clearly described and critically discussed and should be interesting for readers of Molecules. I have only few formal remarks: 1. Please add the list of abbreviations. 2. Numbering of the complex 18 is absent (Table 1 page 6), 3. Please delete the word ones (page 1, line 25), 4. The letters N should be written using italics letter in the chemical names (e.g. page 9, line 280: …2-N-oxide-N,N-dimethyl…). The present manuscript should be acceptable after minor revision for publishing in the Molecules.
Reviewer 3 Report
This is a nice and instructive review of biologically active copper complexes (BACC). It provides an up to date description of the current state of knowledge about cuprocomplexes, including those containing the cuproenzyme SOD1. One aspect that could be improved would consist to raise important questions that remain to be addressed in the field. Comments that might need to be addressed to further enhance the quality for publication are listed below.
- SOD should be written SOD1 throughout the text of the manuscript.
- Lines 25, 38, 39, 75, 76, 107 contain misspelled words. Examples are: ones, nowadeays, tretment, cooper, ions, depend, etc.
- Line 59. The authors mentioned that copper is used as a cofactor by several cuproproteins. Is that so? To the knowledge of the reviewer, there are relatively small number of copper-binding proteins in the human copper proteome (around 54 proteins copper-binding proteins). So, copper-containing proteins are not so abundant in mammals as compared to other metalloenzymes such as Zn- and Fe-dependent proteins.
- In Fig. 1, it is not obvious to find out the description of the illustrated copper-based drugs in the text of the manuscript. It would be important to add a legend for this figure, including a short description for each copper-based compound.
- Title of Table 1… typo : biollogically
- Lines 183 – 194, are there abbreviations of compounds of Table 1? This is unclear.
- What means : functionalised ? This word is found at several places in the text of the manuscript.
Reviewer 4 Report
The manuscript presents a good review of bioactive copper complexes. The work is systematic and well defined. From my point of view, the content is exhaustive and will be an important piece for researchers in this field. As example, the bibliography contains 145 references focused basically on the last 20 years, an they are provided from different field of chemistry. Consequently, I will recommend their acceptance.
Nevertheless, I suggest to revision of the main text, since it must be improved in order to be published. The main text presents small paragraphs containing only one sentence of 2 or 3 lines in which authors provide a basic idea to close a telegram. For this reason, the lack of connection between paragraphs makes a heavy reading, and can make it difficult to disseminate this work. In addition, the use of the English language in some phrase would be much better, even for non-english native speakers.
Another problem I have found is the extensive use of abbreviations in the text. In many cases, they do not appear in the other part of the work. I would recommend a compilation at the ended section to make it easier to read. As an example, I have also detected two different forms for intravenous injection!
And finally, I would be to improve some aspects of Tables: (a) water contents in 2 or 8, (b) X general ligand in 12, (c) R ligand (carboxylato) in 17, (d) undefined ligands in 17, 19-24, ... And also authors would revise the equivalent formulae for complexes due the equivalent species appear with/without parentheses.
These last points should be improved to facilitate a good reading.
Round 2
Reviewer 4 Report
The manuscript presents a good review of bioactive copper complexes, and I would recommend to be published in Molecules. Nevertheless, authors should revise some secondary aspects to provide a better version of this manuscript. I like to provide a model papers to my students, and this presents some remarks that should be avoided.
First, I consider that manuscript contains a large number of small paragraphs, and some of them could probably be fused in only one. As example, I propose the the following, (32->39), (115->121), (169->181), (182->187), (206->215), (254->263), (264->274), (285->284), (310->318), (319->326), (340->348), (349->338), (359->365), (366->373), (381->389), (390->402), (433->445), (446->455), (472->478), (501->510), (563->574), (594->603).
Moreover, other sentence (86->87) should be replaced in the text.
Second, a mistake is found in the formula of line 58, that is written as [Cu(His)2(OH2)2] containing water, but it is missing in figure.
Third, I consider the following mistakes:
* Line 32: “ones” should be removed.
* Line 48: intravenous appears as “i.v.” (and also 195 and 422), but also as “iv” (line 259)
* Line 72: “Kg” should be “kg”, without capital letter.
* Line 180: “Dimethyl” should be “dimethyl”, without capital letter.
* Lines 302, 354 and 501: spaces between chemical name and oxidation state should be removed.
Author Response
see atachment
